# "Are you listening?": Experiences shared online by family caregivers of patients in the palliative phase during the Covid-19-pandemic

Hinke E. Hoffstädt[1]*, Mary-Joanne Verhoef[1], Aranka Akkermans[2], Jenny T. van der Steen[3,4], Arianne Stoppelenburg[1,5], Sita de Vries[6], Everlien de Graaf[6], Saskia C. C. M. Teunissen[6], Iris D. Hartog[1,5], Yvette M. van der Linden[1,5]

1 Center of Expertise in Palliative Care, Leiden University Medical Center, Leiden, The Netherlands, 2 Medical Psychology, Amsterdam University Medical Center, Amsterdam, The Netherlands, 3 Public Health and Primary Care, Leiden University Medical Center, Leiden, The Netherlands, 4 Radboudumc Alzheimer Center and Department of Primary and Community Care, Radboud University Medical Center, Nijmegen, The Netherlands, 5 Netherlands Comprehensive Cancer Organisation (IKNL), Utrecht, The Netherlands, 6 Center of Expertise in Palliative Care Utrecht, Department of General Practice and Nursing Science, Julius Center for Health Sciences and Primary Care, University Medical Center Utrecht, Utrecht, The Netherlands

* h.e.hoffstadt@lumc.nl

**Data Availability Statement:** The doi of the dataset is as follows: doi:10.17026/LS/BWA9ZH. Additionally, the following links both lead to the

## Abstract

### Objectives

In palliative care, it is important for family caregivers to spend time with and care for the patient, and to receive (in)formal support. These elements were compromised during the Covid-19-pandemic. This study investigates what family caregivers of non-Covid-19-patients in the palliative phase shared online during the first wave of the pandemic, and what their communicative intentions were with posting online.

### Methods

To investigate what family caregivers shared online, a reflexive thematic analysis was performed on online newspaper articles and posts on Twitter, Facebook and online forums. To investigate family caregivers' communicative intentions with social media posts, content analysis was conducted guided by Speech Act Theory.

### Results

In total, 412 posts and articles were included, the majority being Tweets (86.7%). Four themes were constructed: 1) 'Being out of touch', 2) 'Fear, worries and uncertainties, 3) 'Disbelief, anger and resistance', and 4) 'Understanding, acceptance and gratitude'. Family caregivers felt overwhelmed with anger, sadness and anxiety, yet some expressing milder or positive attitudes towards the new circumstances. Family caregivers mostly posted online to share their personal experiences and emotions, and to express their opinions about the restrictions.

dataset: https://lifesciences.datastations.nl/dataset.
xhtml?persistentId=doi:10.17026/LS/BWA9ZH.
https://doi.org/10.17026/LS/BWA9ZH.

**Funding:** The project was funded by The
Netherlands Organization for Health Research and
Development (ZonMw; grant numbers 844001804
and 844001706. https://www.zonmw.nl/nl) and the
Jonker Driessen Foundation. The funders had no
role in study design, data collection and analysis,
decision to publish, or preparation of the
manuscript.

**Competing interests:** The authors have declared
that no competing interests exist.

## Conclusions

The pandemic had distinct implications for family caregivers of non-Covid-19-patients in the palliative phase. In future similar circumstances, restrictions should be tailored to individual conditions and interventions should be in place to facilitate contact between patients, family caregivers and healthcare professionals and to support family caregivers.

## Introduction

Being confronted with a life-threatening illness does not only have an enormous impact on patients, but also on the people close to them. Family caregivers provide emotional and physical care for the patient, while also having to deal with their own emotions regarding the patient's deterioration and impending death [1]. Healthcare professionals can help family caregivers deal with these circumstances by providing practical and emotional support, providing information (e.g. regarding the illness, the treatment and prognosis), and to acknowledge them in their role as co-caregiver [2, 3]. These essential elements of palliative care were heavily compromised during the Covid-19-pandemic due to the restrictive measures that were taken to slow spreading of the virus. Furthermore, these measures limited the possibility for family caregivers to spend time with the patient which is known to be a top priority at the end of life for both patients and family caregivers [4–6].

As in other countries, the pandemic severely disrupted healthcare in the Netherlands as treatments and appointments were cancelled or postponed and healthcare teams worked under increased pressure and had to deal with scarcity of personal protective equipment [7–9]. In addition, the Dutch government imposed a partial lockdown in March 2020, which included the closure of nursing homes to visitors. Other healthcare institutions also incorporated visiting restrictions. Daily life was also constrained as the lockdown entailed the strong advice to keep 1,5 meter distance from others, to stay at and work from home, and to only travel when strictly necessary. Furthermore, schools, hospitality facilities and leisure clubs were closed and events were cancelled. The restrictions were gradually loosened in May and June [10].

On an international scale, research has been conducted to understand family caregivers' experiences during the pandemic. Many family caregivers did not get a chance to say goodbye to their loved one and be present when the patient died due to visiting restrictions, resulting in great emotional distress [11, 12]. Contact with healthcare professionals was also limited and largely dependent on telecommunication because of the restrictions and shortage of (time of) healthcare professionals [8, 13]. Family caregivers felt they received insufficient information [14] and were less involved in the care for the patient [8]. During the pandemic, family caregivers' worries about inpatients' emotional wellbeing increased substantially [15]. Family caregivers taking care of community-dwelling patients, in turn, were anxious about the possibility of contracting the virus and infecting their loved one [16, 17]. Furthermore, their caregiving responsibilities increased as home care services were scaled down and support from other family members was limited [7, 18, 19]. Difficulties continued after the patient's death, as only a few people could attend funeral services with limited possible service features [20], and formal and informal grief support were minimal due to lockdowns and the high workload of healthcare professionals [21].

Studies reporting on family caregivers' experiences during the pandemic are diverse, with some focusing on family caregivers of Covid-19-patients [11, 12] or home-dwelling patients

[16, 17, 19], and others focusing specifically on bereavement [20, 21] or end-of-life experiences [12, 14]. However, the perspective of non-bereaved family caregivers of patients with life-threatening diseases other than Covid-19, such as advanced cancer or dementia, has been underrepresented. Therefore, to represent the broad scope of palliative care, this study explores the experiences of bereaved and non-bereaved family caregivers of in- and outpatients with life-threatening diseases other than Covid-19 through an online media search. Online media platforms are a rich source for research data as they are widely used to share a host of information, opinions, thoughts, and feelings [22]. Media searches have been conducted before to investigate experiences of the public during epidemics and pandemics [23, 24]. Furthermore, an analysis of online posts can shed light on family caregivers' communicative intentions with their posts [25–27], which can contribute to a deeper understanding of their needs. As such, this study aims to answer two research questions: 1) What did family caregivers of patients in the palliative phase without Covid-19 share on online media platforms during the first wave of the pandemic in the Netherlands? and 2) What are family caregivers' communicative intentions with their posts on social media platforms?

## Methods

An online media search study was conducted. The Medical Research Ethics Committee of LUMC declared the study exempt from the Medical Research Involving Human Subjects Act (WMO; no. P17.254). The reporting guideline Standards for Reporting Qualitative Research was adhered [28].

### Data collection

A media search was performed on Twitter (known as X as of July 2023), Facebook, online forums and LexisNexis (an online database for newspaper articles) to include posts and articles published during the first wave of the Covid-19-pandemic (March-June 2020) [29]. A list of search terms (S1 File) was constructed by MV, SdV, EdG, ST, YvdL and a member of the patient and family caregiver council of the regional palliative caregiver consortium (Propallia) to include online media posts related to the experiences of family caregivers of patients in the palliative phase. The list was refined by adding commonly encountered relevant terms during the first stages of data collection. Snowball sampling was used to include more posts and articles.

The data were collected manually and in accordance with the terms and conditions of the platforms. Data collection took place in two rounds. At first, data were collected in 2020 by MV and SdV and covered posts and articles from March to May 2020. In 2022, the dataset was expanded by MV and HH to also include posts and articles written in June 2020 to fully cover the first wave of the pandemic. The researchers independently selected relevant articles and posts. Doubts on whether to include a post were discussed among the research group until consensus was reached. Posts and articles were included if they were written by a family caregiver of a non-Covid-19-patient during the first wave of the pandemic in the Netherlands and concerned having a loved one in the palliative phase during the pandemic. Online newspaper articles were also included if they were written by a journalist reporting on the family caregivers' perspective. Posts of bereaved family caregivers were included only if their loved one had died during the first wave. Posts were excluded from analysis if they concerned Covid-19-patients, if the patient was not in the palliative phase, or if the author did not live in the Netherlands. When it was unclear whether exclusion criteria were violated, more posts of the author (if there were any) were examined. When in doubt whether an elderly person was in the palliative phase, MV (elderly care physician resident) assessed patient's vulnerability based

on the available context and the practical guide 'Frail elderly' by the Dutch Patient Safety Program [30]. Posts were excluded if uncertainty persisted regarding any of the exclusion criteria.

Consent from the authors to analyse their posts was not acquired because of the data's public nature.

## Data analyses

**Reflexive thematic analysis.** To explore what Dutch family caregivers shared online during the first wave of the Covid-19-pandemic, an inductive reflexive thematic analysis was performed as this method is suitable to find patterns across datasets [31, 32]. A constructivist-interpretivist approach was applied by MV (elderly care physician resident/researcher in palliative care, PhD) and HH (psychologist/researcher in palliative care, MSc), acknowledging and embracing the influence of their own background and experiences on the coding process and theme development. Besides approaching the data from different perspectives (medical and psychological), the researchers had different experiences related to the first wave of the pandemic that shaped their interpretation of the data. At the time, MV worked as a resident at a nursing home and was therefore, besides being aware of the patients' and family caregivers' perspective, especially aware of the healthcare professionals' perspective. HH had conducted and analysed interviews with (bereaved) family caregivers of Covid-19-patients for a synchronous research project, meaning she had already obtained in-depth understanding of family caregivers' experiences.

MV and HH first familiarised themselves with the data by going through the entire dataset and making notes and exchanging observations while doing so. Next, of each media platform both researchers coded several of the same media posts which they then discussed, developing an initial coding scheme with mostly semantic but also more latent codes. HH coded the remaining data items while keeping an analytic journal in which she noted observations and doubts which were discussed during regular meetings with MV. The coding scheme was adjusted and applied accordingly during the process of coding. When coding was completed, MV and HH first separately gathered thoughts on meaningful patterns of family caregivers' experiences and then developed initial themes during a brainstorm session and follow-up discussions. The themes were further refined by discussing the themes' narratives with the co-authors and by going back and forth between the themes and the raw data. It was aimed for the themes to tell a coherent, meaningful story, each with a unique central organizing concept [31, 32].

**Quantitative content analysis.** To investigate family caregivers' communicative intentions with their social media posts, all parts of posts that were coded thematically were simultaneously coded as part of the quantitative content analysis. Online newspaper articles were excluded from this analysis, as these were written by journalists, making them unsuitable for investigating family caregivers' communicative intentions. A predetermined coding scheme was derived from Weigand's speech act taxonomy [33]. Speech act theory, initially developed by Austin [34], considers peoples' expressions not as mere utterances, but as acts that are performed with an intention, such as advising, warning or suggesting. Searle was the first to develop a taxonomy of such illocutionary acts [35]. This taxonomy was later revised by Weigand as she believed Searle's taxonomy to be insufficiently applicable to real-world interactions [33]. Weigand assumes that people communicate with the intention to come to an understanding with others. Coming to an understanding can take on different forms which she aims to distinguish with her taxonomy. The main categories of Weigand's speech act taxonomy are representatives, directives, exploratives, and declaratives. These categories and their subcategories are further explained in Table 1.

**Table 1. Weigand's taxonomy of illocutionary speech acts [33].**

| Category of illocutionary speech act and explanation | Subcategories | Examples |
|---|---|---|
| **Representatives:** *"To express the world"* <br> With these speech acts the speaker makes a claim to truth. | Subcategories: <br> 1. Representatives as a *simple* claim to truth *("It is so")* <br> a. <u>Assertives</u>: claims to truth which are not immediately evident and have to be proved if the listener asks. <br> b. <u>Identifiers</u>: claims to truth based on knowledge which refer to definitions. <br> c. <u>Informatives</u>: claims that impart new information of which its truth is presupposed. <br> d. <u>Constatives</u>: claims to truth which appear obvious, either relating to the external world or the speaker's inner world without being emotionally affected. <br> e. <u>Emotives</u>: claims implying emotional affect or being overwhelmed by emotions. <br> 2. Representatives as a *modal* claim to truth *("It would/ could/should be so")* <br> a. <u>Conditionals</u>: claims bound to a condition: *"It would be so"* <br> b. <u>Deliberatives</u>: claim expressing the possibility of a state of affairs: *"It could be so"* <br> c. <u>Desideratives</u>: claims expressing desire and include emotional involvement: *"It should be so (1)"* <br> d. <u>Normatives</u>: normative claims: *"It should be so (2)"* | 1.a. "Our economy is going downhill" <br> 1.b. "This image depicts a Eurasian dolphin" <br> 1.c. "On Sunday it's Doris's birthday." <br> 1.d. "Look, the tree's got buds" (external world), "I don't feel well today" (internal world) <br> 1.e. "What a wonderful view!", "What a blasted nuisance!" <br> 2.a. "If we had a different government, things would be better." <br> 2.b. "John might come." <br> 2.c. "If only there were more jobs!" <br> 2.d. "You should be stricter with your daughter." |
| **Directives:** *"To change the world"* <br> With these speech acts the speaker means to bring about a future action or behaviour by the listener. | Subcategories: <br> 1. Claims of an order defined by availability of sanctions <br> 2. Claims of a request on relying on mutual cooperation <br> 3. Claims of a plea appealing to the helpfulness and kindness of the listener | 1. "You're home at 8 o'clock!" <br> 2. "Can you bring a glass of milk, please?" <br> 3. "Would you please extend the deadline with 1 day?" |
| **Exploratives:** *"To ask questions about the world"* <br> With these speech acts, speakers mean to gain information to achieve an understanding of the world. | Subcategories: <br> 1. Knowledge in order to know <br> 2. Knowledge in order to act <br> 3. Knowledge in order to create status functions or to confirm reliability | 1. "What did you do yesterday?", "What's the time?" <br> 2. "What can I do?", "Shall I come to your place?" <br> 3. "Are you telling the truth?", "Are you definitely going to come?" |
| **Declaratives:** *"To create a world"* <br> With these speech acts, speakers declare the existence of a specific state of affairs. This category also includes conventional utterances, as they create what is considered to be civilised behaviour. | No subcategories | "I now pronounce you man and wife", "I name this ship the Queen Elizabeth", "Thank you", "Sorry", etc. |

HH and MV coded 20% of posts on each social media platform separately. Discrepancies between assigned (subcategories) of speech acts were discussed until consensus was reached. Next, HH coded the remaining data items and kept an analytic journal containing doubts which were discussed during regular meetings with MV. The coding scheme was not subject to change during the process. For each post, it was registered which subcategories of illocutionary acts occurred.

Analyses were performed in ATLAS.ti software (versions 9 and 22) [36, 37] and took place between August 2022 and May 2023.

## Results

In total, 412 online media posts and articles were selected for analysis, of which the vast majority were Tweets (86.7%). Most posts related to a patient in a nursing or care home (51.2%) and were written by (or portrayed the perspective of) the patient's adult child (64.5%). More characteristics are summarised in Table 2. First, the results of the thematic analysis are presented, followed by those of the quantitative content analysis. Last, a short paragraph is dedicated to the connection of the results of the two analyses.

**Table 2. Characteristics of online media posts and family caregivers.**

| Characteristics of online media posts, N = 412 | % (n) |
|---|---|
| **Type of online media platform** | |
| Twitter | 86.7% (357) |
| Facebook[1] | 4.9% (20) |
| Online newspaper articles | 4.1% (17) |
| Forum | 4.4% (18) |
| **Date posted/published (2020)** | |
| March | 26.5% (109) |
| April | 17.0% (70) |
| May | 18.0% (74) |
| June | 38.6% (159) |
| **Reaction to another post: yes/no, N = 395[2]** | |
| Yes | 62.8% (248) |
| No | 37.2% (147) |
| Characteristics of family caregivers, N = 251 | % (n) |
| **Gender family caregiver** | |
| Female | 69.7% (175) |
| Male | 26.7% (67) |
| Other | 0.4% (1) |
| Unknown | 3.2% (8) |
| **Patient's place of residence[3]** | |
| Nursing home / care home | 51.2% (129) |
| Home | 20.2% (51) |
| Hospice | 6.7% (17) |
| Hospital | - |
| Unknown or N/A | 21.8% (55) |
| **Relationship of family caregiver to the patient** | |
| Child[4] | 64.5% (162) |
| Partner | 12.7% (32) |
| Niece/nephew | 5.2% (13) |
| Friend | 5.6% (14) |
| Grandchild | 3.2% (8) |
| Sibling[5] | 1.2% (3) |
| Parent | 1.2% (3) |
| Unknown | 6.4% (16) |
| **Number of posts per family caregiver (n)** | |
| 1–5 posts | 97.2% (244) |
| 6–15 posts | 2.4% (6) |
| 15+ posts[6] | 0.4% (1) |

[1] The Facebook search yielded limited results as Facebook changed its search algorithm during data collection.

[2] Not applicable to online newspaper articles.

[3] The total of this category adds up to 252 as one post related to both parents in different healthcare settings.

[4] Includes stepchildren and children-in-law.

[5] Includes stepsibling and sibling-in-law.

[6] Of one family caregiver, 64 items were included.

## What did family caregivers of non-Covid-19-patients in the palliative phase share on online media platforms?

Four themes were constructed that together give an account of what was shared by family caregivers on online media platforms: 1) 'Being out of touch', 2) 'Fear, worries and uncertainties, 3) 'Disbelief, anger and resistance', and 4) 'Understanding, acceptance and gratitude'. The first theme sets the scene of what family caregivers went through during the first wave of pandemic. The following three themes relate to family caregivers' varied emotional responses to the new circumstances.

**1) Being out of touch.** The pandemic and its restrictions put the family caregivers' world upside down. Something fundamental was suddenly jeopardised: being in touch with their ill loved one. For family caregivers of patients residing in healthcare institutions, the visiting restrictions had a large impact on their relationship with the patient. Basic elements of their relationship such as spending time together and physical touch were no longer self-evident. Activities that were especially important as patients were in their last phase of life were out of the question, such as celebrating birthdays or holidays or a last family vacation. Initiatives were introduced to facilitate contact between family caregivers and patients (e.g. video calls and visiting booths with acrylic glass), but many family caregivers were left feeling out of touch nonetheless:

> *"Looking through the window and seeing they are thirsty and in pain. And you're not allowed to help. You can't hold their hand. Even though you are trained in providing care. What a horrible situation to be in. How are you supposed to tell them how much you love them?"* (Twitter)

Even when visiting was possible, this feeling of being out of touch persisted as distance had to be kept and personal protective equipment had to be used:

> *"My mother-in-law was able to see him once after the lockdown. 'I don't know you, who are you?', my father-in-law [who has dementia] asked when he saw his wife entering with a face mask."* (online newspaper)

As the patient entered the dying phase, some family caregivers were allowed to visit to say goodbye. Nevertheless, much valuable time had been lost before that causing the goodbye to feel incomplete:

> *"I haven't been able to see my mother for six weeks. Yesterday, the doctor called me. That I should come without delay. She is dying. She looked at me but couldn't say anything. Six weeks wasted. I will never hear my mother again."* (Twitter)

Others did not get a chance to say goodbye (in person) at all:

> *"I am sad and angry. Sad, because my father is terminal after having had dementia for many years, it's only a matter of hours. I (and the rest of the family) can't be with him to say goodbye because of the crisis that the #coronavirus has put us in. So, this is what saying goodbye to your father looks like now, through a video link."* (Facebook)

Being out of touch with their loved one as they were approaching end-of-life brought about feelings of guilt and falling short:

*"I'm having difficulty sleeping as everything is still running through my mind. I also feel guilty because I was forced to abandon her when she needed me the most."* (Twitter)

Due to the visiting restrictions, family caregivers also became alienated with their role as caregiver. Some were used to actively contribute to the patient's care which was now no longer possible:

*"I looked after my father who has dementia 24/7 for two years. One week before Covid he was admitted to a nursing home. I haven't been allowed to visit him for three weeks, despite having signed up as a volunteer to help with caregiving."* (Twitter)

Many family caregivers were now in the dark regarding the patient's wellbeing. They had to rely on telecommunication with the patient, which for many was impossible due to the patient's cognitive deterioration or digital illiteracy. In some cases, healthcare professionals could provide some information about the patient's health and wellbeing, but for many that could not sufficiently replace seeing it with their own eyes and looking after the patient themselves:

*"I used to visit my husband every day. I would lay out his clothes for the next day and put a handkerchief in his pocket. And now the nurses send me pictures (. . .). And I think: where is his watch?"* (online newspaper)

Last, family caregivers were also out of touch with their social contacts. Some family caregivers wrote about their isolation resulting from the new set of circumstances:

*"I try my best to avoid getting infected, but as a result I have lost all contact with the outside world (friends, choir). It doesn't matter, I'm not complaining, because the most important thing is that HE survives!"* (Forum)

One daughter wrote about the difficulty of not being able to support her mother while visiting her terminally ill husband:

*"When patients are terminally ill, one person may visit once a day. Of course, that person is my mother. My mother whom I can't hold and support while she is at my father's bedside."* (Facebook)

After the patient's death, support was also limited due to restrictions for conducting the funeral (limited amount of people allowed) and the societal lockdown:

*"There you are, crying behind your laptop during the livestream [of the funeral] on YouTube. Covid destroys so much of what you care about."* (Twitter)

As reflected in this theme, family caregivers were out of touch with the patient, both as a loved one and as a caregiver, as well as with their social support system. This resulted in distress and sadness, which is addressed in the following theme.

**2) Fear, worries and uncertainties.** Much of what used to be normal and self-evident became uncertain as the pandemic hit. This brought about many fears and worries for family caregivers. First, the pandemic disrupted care-as-usual for home-dwelling patients. Home care services were downscaled and facilities such as day care and meal services shut down. For

some family caregivers, this resulted in experiencing more responsibility and pressure regarding their caregiving tasks, increasing their burden. Some were hesitant to move their severely care-dependent loved one to a nursing home with visiting restrictions. While taking care of their loved one, family caregivers experienced a great sense of responsibility to protect them from the virus. Limited intensive care capacity added to their concern regarding the consequences if the patient should get infected. Despite being as careful as possible, they could still contract the virus and infect their loved one, potentially causing their death:

*"I am experiencing some mucus in my airways today. This is quite common because of my asthma. But this time it feels like, surely it can't be. . . No fever, no sore throat, it won't be that bad. But I'm getting more and more worried. Also for the people around me. (. . .). Most of all for my mother and stepfather. He is terminal, has cancer, and has had only one lung for years."* (Twitter)

The new set of circumstances made many family caregivers feel insecure regarding the best course of action. Family caregivers expressed doubts on whether they should visit and look after their loved one or instead should stay away to protect them:

*"My father who has dementia lives in a nursing home, so my mother is at home by herself. What do I do? Not visiting my father is painful, but if I don't visit my mother, she might not see anyone for months. A hellish dilemma."* (Twitter)

One family caregiver shared her dilemma whether her frail husband should undergo treatment:

*"I think both scenarios aren't good. Proceeding with the stem cell transplant will obviously make him very vulnerable to Covid-infection and complications, but what will postponing mean for his levels that have been reduced over the past 12 weeks?"* (Forum)

Furthermore, family caregivers worried about their loved one's wellbeing as they felt their loved one was being neglected in the healthcare institution. Some clearly noticed deterioration of the patient's physical and emotional health:

*"I was shocked after my first visit to my father in the nursing home. He looked untended, had long, painful toenails, long hair, and teeth were missing from his lower jaw."* (online newspaper)

Some mentioned improvement of the patient's wellbeing as visitors were allowed again, adding to the conviction that the visiting restrictions had a negative impact on the patient's health:

*"My mother-in-law was 'pronounced' to be in a terminal condition, we could visit her briefly every day. Now she is doing so 'well', that she is no longer terminal, and the visiting restrictions are back in place. Who can make sense of that?"* (Twitter)

Last, many family caregivers worried about the very real possibility of not being able to say goodbye to their loved one. That would mean the patient would die without a loved one by their side. Some family caregivers also worried the patient would misinterpret their absence as disinterest:

*"Sadly, I feel I wasn't there for my wife when she needed me. Who knows what she must have been thinking? 'Where is he?' It wasn't my fault of course, I wasn't allowed in there, but she didn't know that. So sad!"* (Twitter)

In summary, family caregivers experienced many fears and uncertainties which related to the best course of action in caregiving, the patient's wellbeing, and the possibility of not being able to say goodbye. Many family caregivers also experienced disbelief and anger as described in the following theme.

**3) Disbelief, anger and resistance.** Common sentiments reflected by the online media posts were disbelief and indignation as family caregivers felt moral principles were afflicted by the restrictions. How could they be separated from the patient? How could it be acceptable having to make decisions about who could and who could not visit or say goodbye to a loved one or visit their funeral? Many shared the opinion that restrictions should be tailored to individual circumstances, especially when someone is approaching the end of life. Instead, the decision was made *for* them that patients should be protected from the virus no matter what. Meanwhile, many family caregivers felt that the resulting loneliness was more damaging than possibly contracting the virus, even if an infection would prove to be fatal to their loved one:

*"I genuinely wonder whether the remedy isn't worse than the disease. People may live a few months longer, but they are lonelier and more confused. That is some price to pay."* (Twitter)

Furthermore, many family caregivers felt the restrictions were unfair, inconsistent, and did not fit their circumstances, leading some to disregard the recommendations:

*"We choose to be close in his last phase of life, also physically, because that is what my father-in-law needs."* (Twitter)

Several family caregivers were astounded that, for example, booking a flight was possible, yet visiting their loved one was not. Meanwhile, multiple healthcare professionals, some even without personal protective equipment, went to see patients:

*"Can someone explain to me why I, a daughter visiting her father, pose a greater threat than nurse aides and volunteers?"* (Twitter)

Through discussion on the online media platforms, it became clear that restrictions differed between healthcare institutions, adding to the conviction that the restrictions were unfair. Many family caregivers directly addressed the government, the media or healthcare organisations, hoping to activate a new course of action:

*"I want to see my mother, it is truly disgraceful that I can't be close to her!! HUGO [Minister of Healthcare] are you listening?? I don't want to wait another week! I don't care how you fix it, but I want to see her NOW!"* (Twitter)

Family caregivers were also frustrated as they experienced a lack of acknowledgement of their important role in the care for their loved one. They argued that by excluding family caregivers, patients did not receive optimal care. Some family caregivers were even convinced that their loved one would not have died if they would have been able to visit and look after them:

*"I am too tired to fight any longer, but please let the nursing homes provide more customized solutions. Then my wife would still be alive."* (Twitter)

Indignation and anger were also addressed to people ignoring the lockdown recommendations, as those people indirectly put their loved one at danger and contributed to the upholding of restrictions:

*"I haven't seen my terminally ill father for weeks and meanwhile the whole of the Netherlands goes to the beach, [swear word] I hate the Dutch."* (Twitter)

As demonstrated in this theme, family caregivers were struck with disbelief regarding the new circumstances as those conflicted with what they believed to be fair and just. Some expressed more positive views, as reflected in the final theme.

**4) Understanding, acceptance and gratitude.** Family caregivers not only expressed the resistance portrayed in the preceding theme. Some family caregivers considered the restrictions to be understandable, despite their consequences:

*"Her only remaining sister was not allowed to see her after she died. That was very tough. But I do understand. Limit the threat as much as possible for everyone."* (Twitter)

Family caregivers found different ways that helped them to cope with these consequences. One family caregiver shared how she drew support from a higher power. Others described being supported by loved ones or healthcare professionals. Some family caregivers could find comfort in noticing that the patient did not suffer greatly from the restrictions as they did not fully understand what was going on due to their cognitive deterioration. Family caregivers described how they managed to make the most of their last time together, despite the restrictions.

*"On the first of May we got married and, despite Covid, we truly had a wonderful day. My husband said that he forgot he was ill and we really enjoyed the day."* (Forum)

Some were capable to see positive aspects of the restrictions:

*"When my mother died, everyone had to say goodbye individually. In hindsight, this was invaluable, compared to saying goodbye collectively. That's the other side of the coin."* (Twitter)

*"Lockdown gives me peace and space for my grief and sadness, without all kinds of distractions. Being in this bubble with my children feels good, as crazy as that sounds."* (Twitter)

Last, multiple family caregivers expressed gratitude towards healthcare professionals for their hard work during the pandemic and their efforts to facilitate contact and to accommodate their wishes despite the restrictions:

*"Thank you to all nurse aides who are now giving extra attention to our parents on our behalf."* (Twitter)

*"I am very happy how healthcare professionals try to find solutions. We can pick up my mother outside the nursing home so that she can visit my father in the hospice. It's so important now and not something you can make up for later! #grateful."* (Twitter)

In summary, some family caregivers maintained a more positive attitude under the circumstances, some even mentioning silver linings to the restrictions.

## What are family caregivers' communicative intentions with their posts on social media platforms?

In the quantitative content analysis, a total of 1048 speech acts were identified, portraying family caregivers' communicative intentions with social media posts. The distribution between speech act categories is presented in Table 3, including examples for each subcategory. Of all speech acts identified, representative speech acts were by far the most common (81.4%). Most of these were making a simple claim to truth. Informatives were mostly identified when family caregivers shared their factual personal experiences (e.g. *"My foster brother died of pneumonia. No goodbye and a lonely funeral."*) including how they dealt with the circumstances (e.g. *"My dad is incurably ill so we are extremely careful."*). Constatives and emotives were also often identified. With constatives, people shared what was on their mind (e.g. *"I worry about my dad's funeral."*), observations they made (e.g. *"Healthcare professionals try their best."*), and shared the impact of the pandemic on their own wellbeing (e.g. *"I don't sleep well, it's constantly on my mind."*). Emotives were used to share different emotions such as fear, anger (e.g. *"Injustice! Deprivation of liberty of the elderly!"*) and sadness, and also, though less frequently, gratitude, and happiness (e.g. *"It was great to see him after 14 days.")*. Assertives were the least common of the representative speech acts making a simple claim to truth. These were mostly identified when family caregivers expressed their opinion (e.g. *"The Dutch government is currently making mistakes."*). Identifiers did not occur at all.

Representative speech acts making a model claim to truth were regularly identified, though much less often than those making a simple claim to truth. Those speech acts were mostly identified when family caregivers were contemplating how things could have been different

**Table 3. Frequency of speech acts in social media posts.**

| Speech act (N = 1048) | Frequency % (n)[1] | Examples[2] |
|---|---|---|
| Representatives | 92.0% (964) | |
| • as a simple claim to truth | 86.8% (837) | • *"The hospital does not come home for chemotherapy, now we have to use public transport."* (informative)<br>• *"What a sad ending to your life, such a long time without visitors."* (constative)<br>• *"Injustice!", "So unfair, so difficult."* (emotive)<br>• *"With all respect for healthcare providers, they should not be blamed. . ... But our governors. . ."* (assertive) |
| • as a model claim to truth | 13.2% (127) | • *"I may be not even able to call her."* (deliberative)<br>• *"I don't want to go on a flight, I want to go to my mother!"* (desiderative)<br>• *"Birth and death should be spent together!"* (normative)<br>• *"Otherwise, I would have taken the matter to court."* (conditional) |
| Directives | 2.2% (23) | • *"Would this be something for the [name newspaper], [name journalist]? Isn't this becoming more common?"* (request)<br>• *"People, please, let's obey the rules!"* (plea) |
| Exploratives | 2.3% (24) | *"Is anyone experiencing something similar?"* (explorative in order to know) |
| Declaratives | 3.5% (37) | • *"Wishing you strength!"*<br>• *"Thank you!"* |

[1] The frequencies are based on the sum of whether (sub)categories occurred in a post (yes/no), rather than the sum of all speech acts throughout the dataset.

[2] Subcategories of which no example is provided were not identified in the dataset.

(e.g. *"This could have been prevented if I was allowed to visit during the lockdown"*; deliberative) or when they described possible doom scenarios (e.g. *"A second wave will probably mean that I will never get to see my mother again"*; deliberative). Family caregivers expressed their wishes and desires with desideratives (e.g. *"#Iwanthugs*). Normative speech acts entailed statements about how the patient's last phase of life should be spent (e.g. "*To sadness belong hugs. Many hugs."*) as well as critical statements about how healthcare institutions, government or the public should react to and deal with the pandemic (e.g. *A solution must be found for terminally ill people*). Conditional speech acts were rarely identified, but were mostly underpinnings of decisions that family caregivers made (e.g. *"[If I wouldn't cook for my mother daily], she simply would not have food."*).

A small selection of posts urged others to take action with directive speech acts. Those were mostly pleas directed to government to loosen the restrictions or to the public to obey the recommendations and restrictions (*"Take others into consideration and keep distance, stay home!")*. Exploratives were only identified when family caregivers sought information (subcategory explorative in order to know), such as other people's experiences, regarding ambiguous restrictions or by posing critical questions (e.g. *"What will the regulation be for nursing homes where only a few people are infected*? *Will they be closed for forever and a day*?*")*. Declaratives were attributed to conventional utterances, such as thanking others for their support or wishing others well.

## Connection between the two analyses

Generally, each theme comprised a combination of all speech act categories. However, directives are an exception as these were mostly identified in posts related to the theme 'Disbelief, anger and resistance'. This can be explained as this theme reflects family caregivers' wishes for a different political course of action and for the public to behave a certain way. Directives, in turn, are used to prompt a future action or behaviour from the listener. The other three speech act categories were identified across all themes. The vast majority of speech acts being representatives indicates a great need of family caregivers to share their experiences and accompanying emotions. Although the shared experiences were mostly negative as reflected by the themes 'Being out of touch', 'Fear, worries and uncertainties' and 'Disbelief, anger and resistance', some also felt the urge to express a more positive attitude as reflected in the final theme.

## Discussion

This study explored what family caregivers of non-Covid-19-patients in the palliative phase shared on online media platforms during the first wave of the pandemic in the Netherlands and their communicative intentions with social media posts. Family caregivers of patients residing in nursing or care homes shared their experiences with the visiting restrictions and their impact on their relationship with the patient, both as a loved one and as a caregiver. They were angry to be kept at a distance, worried deeply about the patient's wellbeing and worst-case scenarios such as not getting the opportunity to say goodbye. Family caregivers of home-dwelling patients experienced increased care responsibilities while feeling unsure how to protect their loved one from the virus. Meanwhile, social support was limited due to social distancing measures, also during bereavement. Some family caregivers were able to come to peace with the restrictions, even mentioning unexpected positive outcomes, but most expressed indignance and disbelief towards healthcare institutions and the government as an approach tailored to individual circumstances was lacking. Representative speech acts were by far the most common throughout the themes, which indicates that family caregivers mostly posted on social media with the intention to share their personal experiences, emotions, needs

and opinions. Less often, they sought information with explorative speech acts or urged the public or government to a different course of action with directives. Last, social media posts were used to wish others well and thank others for their support using declaratives.

The themes described in this study correspond with results of previous studies reporting on experiences of bereaved family caregivers [12, 20] and family caregivers of home-dwelling patients [16, 17, 19], portraying family caregivers' sadness, anxiety and anger resulting from the pandemic and its restrictions, yet also mentioning more positive attitudes. The current study enhances our understanding of family caregivers' experiences during the whole palliative care trajectory, highlighting their extreme worries about the patient's physical and mental health during their absence. Family caregivers struggled to get information regarding the patients' wellbeing and lacked confidence that the patient was receiving optimal care. The circumstances of the pandemic strike against what family caregivers of terminally ill patients consider to be important, such as being well-informed about the patient's condition, creating a relationship and collaborating with healthcare professionals [38], and possibly the most crucial element of the patient's last phase of life: togetherness [5, 6].

A Dutch interview study performed during the pandemic illustrated that bereaved family caregivers experienced a threefold loss: the loss of a loved one, the loss of the opportunity to say goodbye, and the loss of social support during this difficult time [39]. Our study portrays additional losses that family caregivers went through, even if the patient survived the first wave of the pandemic. They lost invaluable time with the patient as their disease was life-limiting. Furthermore, they lost the opportunity to take care of the patient in their last phase of life, a task that holds great significance for many family caregivers [40]. Considering the above, it is not surprising that studies have shown a negative impact of the pandemic and its restrictions on the grieving process [41, 42].

In this study, it appears that family caregivers did not often use social media to seek information from others with explorative speech acts. This is contradictory to research indicating social media platforms are used in multitude to seek information, also by family caregivers [43, 44]. As the current study relates to the first wave of the pandemic, seeking information may not have been family caregivers' most prominent need. Not only because answers may have been hard to find as the circumstances had been unprecedented, but also because family caregivers felt overwhelmed and felt the strong urge to ventilate their experiences and emotions. This explanation corresponds with the multitude of representatives identified in this study. Furthermore, the vast amount of representatives may be explained as family caregivers were in search for social support [43]. This may have been particularly important during the pandemic with limited physical social support because of social distancing [12, 25]. Directive speech acts, aiming to bring about a future action or behaviour in the reader, were also not commonly identified. However, family caregivers may have hoped to influence the political course of action more implicitly by sharing their poignant experiences through representative speech acts. In doing so, they emphasised the unsustainability of the current state of affairs, as reflected in the theme 'Disbelief, anger and resistance'. They may have hoped to draw attention and receive recognition from healthcare institution executives, politicians and public media platforms as they also engage on social media [45, 46].

## Strengths and limitations

A strength of this study is that it highlights the underrepresented experiences of family caregivers of non-Covid-19-patients in the palliative phase during the Covid-19-pandemic [19]. Second, the speech act analysis offers a new perspective on family caregivers' needs at this time. Third, in contrast to interview data depending on recall, this study gives an account of family

caregivers' real-time emotions and experiences during the first wave of the pandemic [47]. This study also has limitations. First, minimal context of media posts caused for limited depth. Second, not all members of the public (actively) use social media platforms. Social media users tend to be younger and more highly educated [48]. Therefore, the results of this study are not representative of all family caregivers of patients in the palliative phase [49]. Lastly, the authenticity of the social media posts could not be verified [22].

## Implications for practice

This study demonstrates that the needs of family caregivers of non-Covid-19-patients in the palliative phase were severely overlooked during the first wave of the pandemic. The visiting restrictions violated the essential norm and human right to family caregivers of spending time together with the patient at the end of life [50]. In future similar circumstances, government and healthcare organisations should be hesitant to incorporate strict visiting restrictions in their policies as those have a negative impact on patients, family caregivers, and the healthcare professionals [51–53]. Rather, unique visiting plans should be constructed tailored to patient's and family caregivers' needs and what is possible under the current circumstances. Such tailored visiting plans should refrain from treating all potential visitors the same. Different considerations should be made when the visitor is a lifelong partner than when it concerns a more distant friend [50]. When visiting is restricted, extra efforts should be made to make family caregivers feel as involved as possible, to which providing regular updates on the patient's well-being is fundamental [54]. Initiatives set up by intensive care units, such as family support teams, can be taken as an example [55, 56]. Alternatives to visiting that were set up during the pandemic, such as telecommunication and visiting booths, should also be taken into consideration for policy plans. However, as our study and others demonstrate, such alternatives insufficiently replace physical contact [19, 57], and should therefore be regarded as complementary to tailored visitation plans. Lastly, extensive support for family caregivers of both inpatients and home-dwelling patients should be incorporated in healthcare policies, to which services of spiritual counsellors and psychologists can be valuable assets [58, 59]. Such support can also be digital; initiatives such as e-family meeting procedures and telehealth interventions have been implemented during the pandemic with positive results [60, 61].

## Implications for future research

First, since online media posts offer a limited amount of depth, details, and nuance [47], interview studies are needed to better understand the experiences and needs of bereaved and non-bereaved family caregivers of patients in the palliative phase during the pandemic. Second, further research should explore the perspectives of patients, family caregivers, and healthcare professionals regarding the different types of telecommunications that were introduced during the pandemic and the role the widely used social media platforms can have in this. Such understanding can help to develop a strong digital infrastructure [62] which can facilitate contact between the three parties and provision of remote support to family caregivers. Lastly, a better understanding of the determinants of the more positive attitudes of family caregivers during the pandemic can aid the development of supportive strategies.

## Conclusion

The first wave of the pandemic had distinct implications for family caregivers of non-Covid-19-patients in the palliative phase, which were shared in multitude on online media platforms. Family caregivers experienced sadness, worries and anger under the circumstances, yet some also expressed more positive attitudes. In future similar circumstances, restrictions should be

tailored to individual conditions and (digital) interventions should be in place to facilitate contact between patients, family caregivers and healthcare professionals and to support family caregivers.

## Supporting information

**S1 File. Search terms used for data collection.**
(PDF)

**S2 File. Standards for Reporting Qualitative Research (SRQR).**
(DOCX)

## Author Contributions

**Conceptualization:** Mary-Joanne Verhoef, Sita de Vries, Everlien de Graaf, Saskia C. C. M. Teunissen, Yvette M. van der Linden.

**Data curation:** Hinke E. Hoffstädt, Mary-Joanne Verhoef, Sita de Vries.

**Formal analysis:** Hinke E. Hoffstädt, Mary-Joanne Verhoef.

**Funding acquisition:** Saskia C. C. M. Teunissen, Yvette M. van der Linden.

**Investigation:** Hinke E. Hoffstädt, Mary-Joanne Verhoef, Sita de Vries.

**Methodology:** Hinke E. Hoffstädt, Mary-Joanne Verhoef, Aranka Akkermans, Sita de Vries, Everlien de Graaf, Saskia C. C. M. Teunissen, Yvette M. van der Linden.

**Supervision:** Aranka Akkermans, Jenny T. van der Steen, Arianne Stoppelenburg, Everlien de Graaf, Iris D. Hartog, Yvette M. van der Linden.

**Writing – original draft:** Hinke E. Hoffstädt, Mary-Joanne Verhoef, Iris D. Hartog.

**Writing – review & editing:** Hinke E. Hoffstädt, Mary-Joanne Verhoef, Aranka Akkermans, Jenny T. van der Steen, Arianne Stoppelenburg, Sita de Vries, Everlien de Graaf, Saskia C. C. M. Teunissen, Iris D. Hartog, Yvette M. van der Linden.

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
