## [Decision Letter · Decision Letter 0]

9 Jan 2024

PONE-D-23-31483“Are you listening?": Experiences shared online by family caregivers of patients in the palliative phase during the Covid-19-pandemicPLOS ONE

Dear Dr. Hoffstädt,

Thank you for submitting your manuscript to PLOS ONE. After careful consideration, we feel that it has merit but does not fully meet PLOS ONE’s publication criteria as it currently stands. Therefore, we invite you to submit a revised version of the manuscript that addresses the points raised during the review process.

Thank you for your resubmission. Please address the minor revision requests by the reviewers. Please pay particular attention to the call to action section.==============================

We look forward to receiving your revised manuscript.

Kind regards,

Rosemary Frey

Academic Editor

PLOS ONE

Journal Requirements:

Reviewers' comments:

Reviewer's Responses to Questions

**Comments to the Author**

1. Is the manuscript technically sound, and do the data support the conclusions?

Reviewer #1: Yes

Reviewer #2: Yes

2. Has the statistical analysis been performed appropriately and rigorously? 

Reviewer #1: Yes

Reviewer #2: I Don't Know

3. Have the authors made all data underlying the findings in their manuscript fully available?

Reviewer #1: No

Reviewer #2: Yes

4. Is the manuscript presented in an intelligible fashion and written in standard English?

Reviewer #1: No

Reviewer #2: Yes

5. Review Comments to the Author

Reviewer #1: Thank you for the opportunity to review this article. I found it very interesting. My comments centre around having a description of the broader Dutch Govt approach to put some of the findings in context as well as perhaps a sharper discussion integrating the speech act analysis – these are the questions I was left with after reading. There are some other smaller comments/edits. Thank you for doing this study; it is a valuable contribution.

Line 45 - Consistency in naming of pandemic

Line 59 – insert a not?

Line 58 -61 – split the sentence

Line 68 – position of adjective “other” - move to before ‘than’ - same in line 71

Line 89 – A Reference for this proper noun name of a search strategy?

Line 89 – a little description of what was happening in the Netherlands during this first wave wld be helpful. Was this a period of time chosen that reflected a specific period of time the Government approached COVID management in a particular way? Or what changed in June 2020? A bit more context would be helpful to justify this time period chosen – in the background/introduction most likely or somewhere that puts the first theme in context - - or is the situation in lines 47 – 57 Netherlands specific?

Line 157 – a timeframe-period over which this occurred?

Line 164 Prob need to put somewhere the twitter/tweet/X platform changes…

Line 244 – incorrect use of the modal auxiliary “would” – use “should” or change construction to past – “patient became infected”

Line 293 – a reference to the restrictions – would be good to have that broader context in the introduction somewhere as previously described.

Line 385 – Can this summary paragraph reflect the speech act analysis as well, rather than a broad statement

Line 425 – the new perspective by speech act I think needs stronger representation within the discussion – why so many declaratives? Was there a hegemonic situation in The Netherlands which meant explorative or directives were not welcome? Refer back to previous questions about the broader response by the Dutch Govt (lines 89 etc)

Line 467ff – In light of the previous comment, there may be scope to sharpen your discussion and recommendations - especially from the “Call to action” theme, the normative declarative, “Birth and death should be spent together!” and any related directives. See also- Collier A, Balmer D, Gilder E, Parke, R. Patient safety and hospital visiting at the end of life during COVID-19 restrictions in Aotearoa New Zealand: a qualitative study. BMJ Quality & Safety Published Online First: 14 February 2023. doi: 10.1136/bmjqs-2022-015471

Reviewer #2: The front end section clearly set the research context and purpose, and the methods are well-detailed. .

The data is rich and the thematic findings section would benefit from a more nuance organizing structure - for instance, are there subthemes within the main theme? If so signal these in the introduction to each theme. Also, please offer a summarizing statement or short interpretive paragraph at the end of each thematic section. If at all possible, avoid ending a section with a participant's quote. It's good practice to not leave quotations to "speak for themselves".

The section, "a call to action": is this the best title? Is this the authors' view? It seems that in the first part of this section, participants are questioning apparent contradictions within the restrictions. You need to explain how this is a "a call to action". Section 3 and 4 may be the ONE theme - responses to restrictions.

The content analysis results is a stand alone section that needs to be connected to the thematic analysis and discussed in the findings. For instance, how did the content analysis support or not the themes? How can the authors explain the relationship between the two sets of findings. The content analysis results need to be integrated in the discussion more carefully. The final paragraph could be developed further.

The implications sections are sound as is the conclusion.

6. PLOS authors have the option to publish the peer review history of their article (what does this mean?). If published, this will include your full peer review and any attached files.

Reviewer #1: No

Reviewer #2: No

---

## [Author Response · Author response to Decision Letter 0]

23 Feb 2024

Dear editor, 

We highly appreciate the opportunity to submit a revised version of our manuscript ‘“Are you listening?”: Experiences shared online by family caregivers of patients in the palliative phase during the Covid-19-pandemic’. 

We thank the reviewers for their comments and suggestions that contributed to an improved version of the manuscript. Below you can find a point-by-point response to the comments of the two reviewers, in which we also explain how we have addressed the comments. The manuscript with tracked changes shows the revisions we made. 

In the previous version we indicated that we could only make the online newspaper articles publicly accessible. Following the request for broader data availability, we consulted our privacy officer and discussed the options extensively with colleagues. We decided to opt for sharing the Tweet IDs (Twitter) and anonymised Facebook posts. Unfortunately, the forum posts cannot be made available as doing so would violate the terms and conditions of the platforms. All other data have now been submitted for review to the DANS data repository. When accepted for publication the dataset will be available at https://doi.org/10.17026/LS/BWA9ZH. 

Furthermore, to incorporate the feedback of the reviewers we added some references (references 9, 10, 43, 44 and 50). Furthermore, we adjusted reference 39. In the previous version we cited the abstract book of a conference. In the meantime this article has been published and we have cited it accordingly in the new version. Last, we adjusted the formatting of the tables to align with the formatting guidelines of PLOS ONE.

We thank the editor and the reviewers for their efforts and for consideration of the revised version of our manuscript. 

On behalf of all authors,

Your sincerely, 

Hinke E. Hoffstädt.

#Reviewer 1

“Thank you for the opportunity to review this article. I found it very interesting. My comments centre around having a description of the broader Dutch Govt approach to put some of the findings in context as well as perhaps a sharper discussion integrating the speech act analysis – these are the questions I was left with after reading. There are some other smaller comments/edits. Thank you for doing this study; it is a valuable contribution.”

Response: We thank the reviewer for the kind words and taking the time to read our manuscript, identifying language errors and places that need clarification or improvement. Please find below our responses and the adjustments we made to the manuscript. The line numbers in our responses refer to the clean version of the revised manuscript.

1. “Line 45 - Consistency in naming of pandemic”

Revision: We changed ‘-19-pandemic’ to ‘Covid-19-pandemic’ (line 45)

2. “Line 59 – insert a not?” and “Line 58-61 – split the sentence”

Revision: We split and revised the sentence to improve clarity: “Family caregivers taking care of community-dwelling patients, in turn, were anxious about the possibility of contracting the virus and infecting their loved one. Furthermore, their caregiving responsibilities increased as home care services were scaled down and support from other family members was limited.” (lines 65-68)

3. “Line 68 – position of adjective “other” - move to before ‘than’ - same in line 71”

Revision: We adjusted both sentences as suggested by the reviewer. (lines 75 and 78)

4. “Line 89 – A Reference for this proper noun name of a search strategy?”

Response: The Domain, Determinant and Outcome-outline is an alternative search strategy to the more well-known PICO model (population, intervention, comparison, outcome), commonly used in the Netherlands. However, we realize now that the outline is less well-known on an international scale. Considering the mentioning of the DDO-outline is not necessary to explain our search strategy, we decided to omit this term and adjust the paragraph accordingly: “A media search was performed on Twitter (known as X as of July 2023), Facebook, online forums and LexisNexis (an online database for newspaper articles) to include posts and articles published during the first wave of the Covid-19-pandemic (March-June 2020).[29] A list of search terms (S1) was constructed by MV, SdV, EdG, ST, YvdL and a member of the patient and family caregiver council of the regional palliative caregiver consortium (Propallia) to include online media posts related to the experiences of family caregivers of patients in the palliative phase. The list was refined by adding commonly encountered relevant terms during the first stages of data collection. Snowball sampling was used to include more posts and articles.” (lines 93-100)

5. “Line 89 – a little description of what was happening in the Netherlands during this first wave wld be helpful. Was this a period of time chosen that reflected a specific period of time the Government approached COVID management in a particular way? Or what changed in June 2020? A bit more context would be helpful to justify this time period chosen – in the background/introduction most likely or somewhere that puts the first theme in context - - or is the situation in lines 47 – 57 Netherlands specific?”

Response: We agree with the reviewer that the context of the Covid-19-pandemic in the Netherlands specifically was insufficiently portrayed in the previous version of the manuscript. 

Revision: We now added information in the introduction to provide more context of what was happening in the Netherlands during March-June 2020 and why this period was selected as time frame: “As in other countries, the pandemic severely disrupted healthcare in the Netherlands as treatments and appointments were cancelled or postponed and healthcare teams worked under increased pressure and had to deal with scarcity of personal protective equipment.[7-9] In addition, the Dutch government imposed a partial lockdown in March 2020, which included the closure of nursing homes to visitors. Other healthcare institutions also incorporated visiting restrictions. Daily life was also constrained as the lockdown entailed the strong advice to keep 1,5 meter distance from others, to stay at and work from home, and to only travel when strictly necessary. Furthermore, schools, hospitality facilities and leisure clubs were closed and events were cancelled. The restrictions were gradually loosened in May and June.[10]” (lines 49-57) 

6. “Line 157 – a timeframe-period over which this occurred?”

Revision: In the sections data collection (lines 101-105) and data analyses (lines 167-168) we specified when the research activities took place.

7. “Line 164 Prob need to put somewhere the twitter/tweet/X platform changes…”

Revision: We added ‘known as X as of July 2023’ in parentheses after the first mention of Twitter in line 93. 

8. “Line 244 – incorrect use of the modal auxiliary “would” – use “should” or change construction to past – “patient became infected””

Revision: We changed “would” to “should” as suggested by the reviewer. (line 264)

9. “Line 293 – a reference to the restrictions – would be good to have that broader context in the introduction somewhere as previously described.”

Revision: We added some of the context of the first wave of the pandemic in The Netherlands in the introduction as indicated in our response to comment no.5. 

10. “Line 385 – Can this summary paragraph reflect the speech act analysis as well, rather than a broad statement”

Revision: We can see your point. Rather than the broad statement at the beginning of the discussion, we now ended the first, summarizing paragraph of the discussion with a more specific description of the results of the speech act analysis: “Representative speech acts were by far the most common throughout the themes, which indicates that family caregivers mostly posted on social media with the intention to share their personal experiences, emotions, needs and opinions. Less often, they sought information with explorative speech acts or urged the public or government to a different course of action with directives. Last, social media posts were used to wish others well and thank others for their support using declaratives.” (lines 436-441)

11. “Line 425 – the new perspective by speech act I think needs stronger representation within the discussion – why so many declaratives? Was there a hegemonic situation in The Netherlands which meant explorative or directives were not welcome? Refer back to previous questions about the broader response by the Dutch Govt (lines 89 etc)”

Response: We believe the reviewer may have meant ‘why so many representatives’ instead of ‘declaratives’. However, the point of the reviewer still stands as not many explorative and directive speech acts were identified which we had hardly addressed in the discussion section. 

Revision: We elaborated the last paragraph of the first section of the discussion, now offering explanations on why only a limited number of exploratives and directives were identified: “In this study, it appears that family caregivers did not often use social media to seek information from others with explorative speech acts. This is contradictory to research indicating social media platforms are used in multitude to seek information, also by family caregivers.[43,44] As the current study relates to the first wave of the pandemic, seeking information may not have been family caregivers’ most prominent need. Not only because answers may have been hard to find as the circumstances had been unprecedented, but also because family caregivers felt overwhelmed and felt the strong urge to ventilate their experiences and emotions. This explanation corresponds with the multitude of representatives identified in this study. Furthermore, the vast amount of representatives may be explained as family caregivers were in search for social support.[43] This may have been particularly important during the pandemic with limited physical social support because of social distancing.[12,25] Directive speech acts, aiming to bring about a future action or behaviour in the reader, were also not commonly identified. However, family caregivers may have hoped to influence the political course of action more implicitly by sharing their poignant experiences through representative speech acts. In doing so, they emphasised the unsustainability of the current state of affairs, as reflected in the theme ‘Disbelief, anger and resistance’. They may have hoped to draw attention and receive recognition from healthcare institution executives, politicians and public media platforms as they also engage on social media.[45,46]” (lines 462-478).

12. “Line 467ff – In light of the previous comment, there may be scope to sharpen your discussion and recommendations - especially from the “Call to action” theme, the normative declarative, “Birth and death should be spent together!” and any related directives. See also- Collier A, Balmer D, Gilder E, Parke, R. Patient safety and hospital visiting at the end of life during COVID-19 restrictions in Aotearoa New Zealand: a qualitative study. BMJ Quality & Safety Published Online First: 14 February 2023. doi: 10.1136/bmjqs-2022-015471”

Response: We agree with the reviewer that the crucial importance of family caregivers spending time with the patient at the end of life could be made more explicit in the recommendations. Furthermore, we believed the reference suggested by the reviewer to be a valuable study to support this point. 

Revision: We adjusted the section ‘Implications for practice’ to emphasise the importance of family caregivers and patients spending time together. The sentences that were added based on the reviewer’s comment are made bold: “This study demonstrates that the needs of family caregivers of non-Covid-19-patients in the palliative phase were severely overlooked during the first wave of the pandemic. The visiting restrictions violated the essential norm and human right to family caregivers of spending time together with the patient at the end of life.[50] In future similar circumstances, government and healthcare organisations should be hesitant to incorporate strict visiting restrictions in their policies as those have a negative impact on patients, family caregivers, and the healthcare professionals.[51-53] Rather, unique visiting plans should be constructed tailored to patient’s and family caregivers’ needs and what is possible under the current circumstances. Such tailored visiting plans should refrain from treating all potential visitors the same. Different considerations should be made when the visitor is a lifelong partner than when it concerns a more distant friend.[50]” (lines 491-500)

#Reviewer 2

1. “The front end section clearly set the research context and purpose, and the methods are well-detailed.”

Response: We thank the reviewer for the time and effort to read our manuscript, pointing out its strengths and where it can be improved. Please find below our response to the comments and the revisions in the manuscript. The line numbers in our responses refer to the clean version of the manuscript.

2. “The data is rich and the thematic findings section would benefit from a more nuance organizing structure - for instance, are there subthemes within the main theme? If so signal these in the introduction to each theme.”

Response: We agree with the reviewer that the results section benefits from more structure. According to Braun & Clarke, one should be careful with the use of subthemes in thematic analyses as these may interfere with the coherence of the narrative. 

Revisions: We rearranged some elements within the first theme to structure the narrative. It has been rearranged so that related topics are closer together (starting with not being able to simply be together with the patient, also nearing their death, followed by being out of touch with their role as caregiver and finishing the theme with the notion of being out of touch with the support system). (lines 212-254)

3. “Also, please offer a summarizing statement or short interpretive paragraph at the end of each thematic section. If at all possible, avoid ending a section with a participant's quote. It's good practice to not leave quotations to "speak for themselves".”

Revisions: At the end of each theme, we added a summarizing sentence to prevent a theme from ending with a quote. Furthermore, this helped to better indicate the relationship between the themes. 

Lines 252-254 (theme 1): “As reflected in this theme, family caregivers were out of touch with the patient, both as a loved one and as a caregiver, as well as with their social support system. This resulted in distress and sadness, which is addressed in the following theme.”

Lines 296-299 (theme 2): “In summary, family caregivers experienced many fears and uncertainties which related to the best course of action in caregiving, the patient’s wellbeing, and the possibility of not being able to say goodbye. Many family caregivers also experienced disbelief and anger as described in the following theme.” 

Lines 338-340 (theme 3): “As demonstrated in this theme, family caregivers were struck with disbelief regarding the new circumstances as those conflicted with what they believed to be fair and just. Some expressed more positive views, as reflected in the final theme.”

Lines 367-368 (theme 4): “In summary, some family caregivers maintained a more positive attitude under the circumstances, some even mentioning silver linings to the restrictions.” 

4. “The section, "a call to action": is this the best title? Is this the authors' view? It seems that in the first part of this section, participants are questioning apparent contradictions within the restrictions. You need to explain how this is a "a call to action".” 

Response: We agree with the reviewer that ‘a call to action’ may not have been the best choice for the third theme as not all elements described here clearly relate to ‘a call to action’. 

Revision: We changed the theme’s title to ‘Disbelief, anger and resistance’, more in line with the second and fourth theme. However, as this causes the title of the first theme to stand out from the others, we dedicated two sentences at the beginning of the results section to explain the relationship between the four themes to help understand why the fi

---

## [Decision Letter · Decision Letter 1]

5 Sep 2024

“Are you listening?": Experiences shared online by family caregivers of patients in the palliative phase during the Covid-19-pandemic

PONE-D-23-31483R1

Dear Dr. Hoffstädt ,

We’re pleased to inform you that your manuscript has been judged scientifically suitable for publication and will be formally accepted for publication once it meets all outstanding technical requirements.

Kind regards,

Rosemary Frey

Academic Editor

PLOS ONE

Additional Editor Comments (optional):

Reviewers' comments:

Reviewer's Responses to Questions

**Comments to the Author**

1. If the authors have adequately addressed your comments raised in a previous round of review and you feel that this manuscript is now acceptable for publication, you may indicate that here to bypass the “Comments to the Author” section, enter your conflict of interest statement in the “Confidential to Editor” section, and submit your "Accept" recommendation.

Reviewer #1: All comments have been addressed

2. Is the manuscript technically sound, and do the data support the conclusions?

Reviewer #1: Yes

3. Has the statistical analysis been performed appropriately and rigorously? 

Reviewer #1: I Don't Know

4. Have the authors made all data underlying the findings in their manuscript fully available?

Reviewer #1: Yes

5. Is the manuscript presented in an intelligible fashion and written in standard English?

Reviewer #1: Yes

6. Review Comments to the Author

Reviewer #1: Thank you for your thoughtful feedback and changes to your manuscript which I thought addressed my review adequately.

7. PLOS authors have the option to publish the peer review history of their article (what does this mean?). If published, this will include your full peer review and any attached files.

Reviewer #1: No

---

## [Editor Report · Acceptance letter]

16 Sep 2024

PONE-D-23-31483R1 

PLOS ONE

Dear Dr. Hoffstädt, 

I'm pleased to inform you that your manuscript has been deemed suitable for publication in PLOS ONE. Congratulations! Your manuscript is now being handed over to our production team.

Kind regards, 

on behalf of

Dr. Rosemary Frey 

Academic Editor

PLOS ONE